# Behavioral Studies of p62 KO Animals with Implications of a Modulated Function of the Endocannabinoid System

**DOI:** 10.3390/cells11091517

**Published:** 2022-04-30

**Authors:** Christina Keller, Sebastian Rading, Laura Bindila, Meliha Karsak

**Affiliations:** 1Neuronal and Cellular Signal Transduction, Center for Molecular Neurobiology Hamburg (ZMNH), University Medical Center Hamburg-Eppendorf, 20246 Hamburg, Germany; christina.kroos@zmnh.uni-hamburg.de (C.K.); sebastian.rading@zmnh.uni-hamburg.de (S.R.); 2Clinical Lipidomics Unit, Institute of Physiological Chemistry, University Medical Center of the Johannes Gutenberg University Mainz, 55128 Mainz, Germany; bindila@uni-mainz.de

**Keywords:** cannabinoid receptor, CB1, SQSTM1, LC-MS/MS, arachidonic acid, elevated plus maze, light–dark box, hypolocomotion

## Abstract

Elementary emotional states and memory can be regulated by the homeostasis of the endocannabinoid system (ECS). Links between the ECS and the autophagy receptor p62 have been found at the molecular level and in animal studies. This project aimed to validate the anxiety and memory phenotype of p62 knockout (KO) animals and whether the ECS plays a role in this. We examined the behavior of p62 KO animals and analyzed whether endocannabinoid levels are altered in the responsible brain areas. We discovered in age-dependent obese p62 KO mice decreased anandamide levels in the amygdala, a brain structure important for emotional responses. Against our expectation, p62 KO animals did not exhibit an anxiety phenotype, but showed slightly increased exploratory behavior as evidenced in novel object and further tests. In addition, KO animals exhibited decreased freezing responses in the fear conditioning. Administration of the phytocannabinoid delta^9^-tetrahydrocannabinol (THC) resulted in lesser effects on locomotion but in comparable hypothermic effects in p62 KO compared with WT littermates. Our results do not confirm previously published results, as our mouse line does not exhibit a drastic behavioral phenotype. Moreover, we identified further indications of a connection to the ECS and hence offer new perspectives for future investigations.

## 1. Introduction

Biochemical studies represent critical approaches to identify molecular interactions and thus identify functional protein networks. To further understand such molecular complexes at a physiological level, in vivo studies help to gain insights and to determine functional consequences. Molecular interaction of the endocannabinoid system (ECS) with the protein p62 has previously been shown in autophagy processes [1,2,3], in biochemical protein–protein interaction studies [4], and in bone formation and remodeling in mice [5]. The link between the ECS and p62 provides a putative molecular connection for the interplay between the two systems. It may also be important for treating neurological diseases in which both systems are implicated.

The ECS offers excellent pharmacological targets for neuroprotection and for anxiety disorders due to its multiple mechanisms of action [2]. This endogenous system is composed of the G-protein coupled cannabinoid receptors CB1 and CB2 and their lipid ligands 2-arachidonoylglycerol (2-AG) and anandamide (AEA). In brain regions involved in regulating mood and emotion, such as the prefrontal cortex, hippocampus, amygdala, and hypothalamus, the CB1 receptor is expressed, and both endocannabinoids (eCBs) are biosynthesized [6,7]. Studies in transgenic mice lacking CB1 receptors showed increased anxiety, stress, and fear responses along with depression-like behaviors [8]. CB1 KO mice exhibited anxiogenic behavior in the light–dark box test [8] and reduced entries and time spent in the open arms of the elevated plus-maze [9]. In line with that, mice with a deletion of the 2-AG synthesizing enzyme diacylglycerol lipase (DAGL) showed a reduced exploration of the central arena in an open-field test and increased anxiety-related behavior in the light–dark box [10].

Pharmacological blockade of the endocannabinoid-degrading enzymes monoacylglycerol lipase (MAGL) and fatty acid amide hydrolase (FAAH) increased brain 2-AG and AEA, respectively, and produced anxiolytic effects in rats [11]. In addition, an endocannabinoid reuptake inhibitor revealed anxiolytic effects in the basolateral amygdala in mice [12]. The CB1 receptor inverse agonist rimonabant SR141716A elicited anxiogenic behavior in rats [11]. The drug rimonabant was approved to treat obesity, but patients developed anxiety- and depression-related symptoms [13], which led to the discontinuation of its admission.

Specifically, in the basolateral amygdala, eCB signaling has been linked to the regulation of stress response and anxiety. Administration of 2-AG and anandamide reuptake inhibitors into the amygdala produced anxiolytic effects through the CB1 receptor expressed in glutamatergic synapses [12,14,15,16]. Furthermore, acute stress has been shown to decrease AEA levels in the amygdala and hippocampus of mice and rats [17]. However, previous studies showed that the ECS regulatory mechanisms are complex and not always intuitively evident. For example, a substance can have both anxiogenic and anxiolytic effects via biphasic processes [18]. Furthermore, the pharmacological elevation of AEA in the basolateral amygdala resulted in different responses that depended on the emotional state of the rats. Under conditions of low emotional arousal, the elevation of AEA decreased anxiety. Under conditions of high emotional arousal, the pharmacological elevation of AEA induced anxiety [17,19]. Thus, the endocannabinoid system, including the CB1 receptor, endocannabinoids, and enzymes involved in their degradation and formation, appears to play an essential role in regulating anxiety, stress response, and depression through very complex processes. Furthermore, the ECS modulates emotional responses and memory formation and retrieval [18,20].

Several neurological functions have also been identified for the protein p62. The multidomain protein serves as a signaling hub for several pathways and is associated with neurodegeneration in humans [21]. Brain samples from patients with neurodegenerative diseases contain large amounts of misfolded and abnormally aggregated proteins [21]. In addition, substances targeting p62 to induce autophagy have promoted the degradation of unwanted molecules. The protein p62 can bind ubiquitinated tau protein and transport it for proteasomal degradation [22]. Genetic inactivation of p62 leads to Alzheimer’s disease-like conditions, including Aβ accumulation, tau hyperphosphorylation, and neurodegeneration in mice [23]. Working memory and spatial memory of p62 KO and WT mice indicated neurodegeneration in mature and obese (6 months old) p62 KO mice, as indicated by the loss of synapses and defect in working memory in the water maze, 8-arm radial maze, and T-maze [23]. Such effects of neurodegeneration were not observed in nonobese, 2-month-old p62 KO mice, suggesting that tau alternation and neurodegeneration in mice are affected by age [23]. The same group described in mature p62 KO mice increased anxiety, decreased exploratory behavior, and showed signs of depression [23]. Complementarily, improvement in spatial learning and long-term memory was observed in mice overexpressing p62 in neural tissue (OEp62 mice) [24]. In addition, OEp62 mice showed increased mitochondrial energy output [25], one of the many causes of anxiety disorders [26]. OEp62 mice showed reduced anxiety, suggesting that p62 may be a suitable target for treating anxiety disorders [24]. 

In the context of the current situation of high data generation and the “reproducibility crisis”, independent validations of published results are also important in the field of behavioral biology [27]. Particularly in analyses of genetically modified mouse models, which are not available to the general public, such studies and phenotyping of, e.g., knockout mice are lacking. In our project, we analyzed a p62 KO mouse provided by the Knockout Mouse Programme (KOMP) repository for scientific purposes. Our work aimed to determine whether these knockout mice also exhibit altered anxiety behavior and memory deficits and whether we can identify a link to the ECS. Thus, we aimed to analyze the behavior of p62 KO animals and compare them with their age-matched WT littermates in terms of anxiety and exploratory behavior and working, long-term, and fear memory. To determine whether the ECS is molecularly connected to the phenotype of p62 KO mice, we also analyzed the levels of AEA and 2-AG in specific brain regions and treated the mice with THC. Our results showed modulated endocannabinoid levels, signs of slightly increased exploratory behavior, and reduced freezing responses. 

## 2. Materials and Methods

### 2.1. Experimental Animals

Knockout-first p62 mice (C57BL/6N-Sqstm1tm1a(KOMP)Wtsi) were available from the KOMP directory (UC Davis, University of California) on a C57BL/6N background (ID: 41073) and carried a promoter-driven selection cassette (lacZ and neomycin). Before analysis, these mice were crossed with C57BL/6J mice (Charles River) for >6 generations to produce fertile offspring that grew normally. All mice were kept in a 12 h:12 h light–dark cycle (light on at 6 p.m.), with a room temperature of 22 °C and 55% humidity. Animals were housed alone or in groups of no more than 5 mice per cage in standard Makrolon type II (22 cm × 16 cm × 14 cm) cages with bedding, nesting material, and ad libitum access to food and water. All mice were kept and tested according to the German and European Community laws on the protection of experimental animals and were approved by the Behörde für Gesundheit und Verbraucherschutz of the City of Hamburg (project identification code number 154_16; date of approval 14 July 2016).

### 2.2. Behavior in General

Mice were transferred at least 30 min prior to the start of the experiment to a room next to the behavior room. The room was illuminated with red light and was separated by a door from the behavior room. The material and arenas were cleaned with 70% ethanol between each subject. All tracks and the analysis used in an open field, elevated plus maze, light–dark box, Y-maze, and novel object recognition were conducted using the tracking software EthoVision (XT 6.1, Noldus, Wageningen, The Netherlands).

### 2.3. Open Field

The open field test was based on rodents’ natural aversion to open spaces. When exploring the empty arena, mice tended to avoid the center and stay closer to the walls. The open field test was used to examine locomotion, exploratory, and anxiety-related behaviors. The arena was made from white non-reflecting plastic and measured 49 cm × 49 cm × 49 cm. Walls (28 cm) surrounded the arena. The mouse was introduced to the arena by placing it carefully at the center. Each mouse had 10 min time to explore the arena freely. The arena was divided into 2 different zones by using the tracking software EthoVision (XT 6.1, Noldus, Wageningen, The Netherlands): the border zone (49 cm × 49 cm) and the center zone (25 cm × 25 cm). The time spent in each zone and the distance traveled were examined.

### 2.4. Elevated Plus Maze

Mice avoided open arms to be protected against predators. This trait was used in the elevated plus maze to study anxiety and exploratory behavior. Mice were introduced into the center (5 cm × 5 cm) of the plus-shaped platform facing a closed arm. Mice had 5 min time to explore the arena freely. The open and closed arms had a length of 30 cm × 5 cm and the platform was 60 cm above the ground. The total distance mice traveled and the time spent exploring the closed and open arms were measured.

### 2.5. Light–Dark Box

Rodents favored dark and closed spaces over bright and open arenas. The light–dark box test was chosen to examine the anxiety and exploratory behavior of mice. The arena consisted of an open and bright illuminated arena (450 Lux, 30 cm × 20 cm) and a closed dark compartment (15 cm × 20 cm) that was surrounded by 20 cm high walls. Both arenas were connected by a small hole in the dark compartment that allowed mice to enter both arenas. Mice were introduced into the arena by placing them directly in front of the hole to the dark compartment. Mice directly entered the dark compartment and the time that mice needed to re-enter the light arena was measured. Additionally, the total distance traveled within the open arena, as well as the total time spent in each arena, were analyzed. Mice had 5 min to explore both arenas.

### 2.6. Y-Maze

The Y-Maze test was sensitive to analyzing working memory, spontaneous alternation, and locomotor activity of the mice. The Y-shaped testing arena (35 cm × 8 cm × 30 cm) consisted of 3 arms at a 120° angle from each other. The arms were named A, B, and C. The starting position for all mice was arm A. Mice were free to explore all arms for a total of 5 min. A working memory error was counted when a mouse entered a recently visited arm. Therefore, a sequence of A B C B A B was counted as 2 working memory errors. The number of errors as well as the number of correct choices was calculated during 5 min of testing time.

The total number of arms visited, alternation behavior describing the percentage of correct choices of arm entries, and the distance moved were additionally analyzed.

### 2.7. Novel Object Recognition Test

The novel object recognition test used the natural curiosity of mice for novel objects to investigate long-term memory. In order to focus attention on the objects, mice were introduced to the empty open field arena on 2 consecutive days (for each 10 min), leading to a gradual decrease in exploratory activity referred to as habituation. On the next day, during the acquisition phase, mice had 10 min time to explore 2 identical objects placed in the open field arena. Analysis was conducted using the tracking software EthoVision (XT 6.1, Noldus, Wageningen, The Netherlands). The software tracked the nose tip of the mouse and measured the time spent with its nose in a 4 cm wide zone that was drawn around each object. Long-term memory was tested after 24 h when mice were reintroduced to the open field arena containing one familiar object and a novel object. To reduce preferences towards one kind of object or location, objects were counterbalanced.

### 2.8. Pole Test

This test was chosen to analyze motor coordination, balance, and motor learning of mice. A vertical pole (48 cm long and 0.8 cm of diameter) made from rough wood was attached to a board. A mouse was placed at the top of the rod, grasping the pole with all four paws and facing upwards. This apparatus was placed into the home cage of the mouse, in order to motivate the mouse to climb down the pole. In favor of reaching the home cage, mice needed to turn 180° and climb down the pole with the head pointing downwards. The earliest point for the mouse to turn was called level 1 (above 32 cm), followed by level 2 (between 32 cm and 16 cm), and at the bottom level 3 (below 16 cm). The time mice needed to reach their home cage and the level to turn were evaluated. The test was carried out on 3 consecutive days with 3 trials a day.

### 2.9. Rotarod

Sensorimotor function, coordination, strength, and endurance can be detected by the rotarod test (TSE Systems: Jones and Roberts, Accelerated Rota-Rod for mice 7650). Mice underwent 12 trials of running during a 3-day period of training. Between trials, mice were given at least 30 min of rest. Trials were completed in acceleration mode for 5 min. Rotation speed increased from 4 to 40 rpm during the first 4 min. If a mouse fell, the latency to fall was recorded. After 5 min of running, mice were picked up and brought back to their home cages.

### 2.10. Fear Conditioning

Classic conditioning after Pavlov uses a neutral stimulus and pairs it with an unconditioned stimulus that will provoke an unconditioned response. Following conditioning, the unconditioned response will be triggered by the neutral stimulus. In this laboratory paradigm, specific environmental cues and a tone as a neutral stimulus were paired with an aversive stimulus using a foot shock as an unconditioned stimulus that will lead to freezing as an unconditioned response. Thereafter, the previously neutral stimulus (environment/tone) acquires an aversive property (freezing response) through associative learning. Fear conditioning allowed us to study fear, aversive learning, and memory of mice.

Conditioning phase: mice were placed into the TSE Multi-Conditioning System (TSE Systems, Bad Homburg, Germany). A floorless, transparent, and rectangular arena (20 cm × 23 cm × 35 cm) was placed on to the metallic grid floor where the food shock was delivered. A removable lid prevented mice from jumping off the arena. Other environmental cues were bright light (50 Lux), constant background noise (80 dB), and the smell of 70% ethanol that was used for cleaning. Mice were free to explore the arena for 1.5 min. Afterward, a tone was played for 30 s and during the last 2 s a foot shock of 0.5 mA was delivered. This protocol was repeated 2 more times and the percentage mice spent freezing was analyzed.

Contextual memory: on the following day, mice were introduced into the same setup as used in the conditioning phase. This time no tone and foot shock were applied. The time mice spent freezing was measured during a period of 5 min.

Cued memory: different environmental cues were used. The arena was rectangular (20 cm × 23 cm × 35 cm) and non-transparent except for one side. The transparent side was placed in front of a black star that was attached to the wall of the TSE Multi-Conditioning System. The grid floor was covered with a black acrylic board and new wooden bedding material. An additional board was diagonally placed into the arena to create a triangular-shaped space. Dim red light (5 Lux), minimal background noise and 1% acetic acid were used. Cued fear memory was examined 2 h after the contextual memory test and was repeated 7 days later.

### 2.11. THC Experiment

The body temperature of each female mouse (aged 7–8 months) was measured with an infrared thermometer (Raynger ST, Raytek). The red dot of the thermometer was directed under the sternum of each mouse and was measured with a distance of 3 to 4 cm. The body temperature was measured on the day before the start of the experiment and directly before the start of the experiment. Both measurements were taken together and used as a baseline. Mice were injected with 5 mg/kg of THC (in DMSO:Tween 80:0.9% NaCl-solution in a ratio of 1:1:18) using a 1 mL syringe and a G27 (Sterican 0.40 × 20 mm) cannula for intraperitoneal injection. After injection, the mouse was placed in a new cage equipped with old bedding and nesting material. Then, 30 min after injection, the body temperature was measured, and the mouse was placed in the arena of an open field for 10 min. THC (Dronabinol) was obtained from THC Pharm GmbH, Frankfurt/Main, Germany.

### 2.12. Endocannabinoid Measurement 

Endocannabinoid measurement was performed in brain tissues that were freshly isolated from p62 KO and WT animals. The samples were snap-frozen in liquid nitrogen and stored at −80 °C until further use. Endocannabinoids and arachidonic acid (AA) were extracted from tissues as previously described. Briefly, for the extraction [28,29], internal standard mixture of deuterated eCBs, ethyl acetate/n-hexane (9:1, *v*/*v*) serving as extraction solvent, and 0.1% formic acid serving as homogenization solvent were added to the samples and quickly vortexed, followed immediately by homogenization in a tissue lyzer for 1 cycle of 30 s at 30 Hz. Next, samples were centrifuged (10 min; 16,000× *g*; 4 °C) and kept at −20 °C for 10 min. The upper organic phase was withdrawn and transferred to 96-well plates. The aqueous phase was further used for protein content determination using a bicinchoninic acid (BCA) assay. 

After evaporation to dryness of the organic phase, the extract was reconstituted in 50 μL acetonitrile/H_2_O (1:1, *v*/*v*) for Liquid Chromatography (LC)/Multiple Reaction Monitoring (MRM) analysis. The solution of extracted endocannabinoids (20 μL) was injected and separated on a Phenomenex Luna 2.5 μm C18(2)-HST column, 100 × 2 mm^2^, combined with a pre-column (C18, 4 × 2 mm^2^; Phenomenex, Aschaffenburg, Germany). For LC separation, acetonitrile containing 0.1% formic acid, serving as solvent B, was increased over 2 min from 55 to 90% and maintained at 90% for 5.5 min, where solvent A was 0.1% formic acid. The separated endocannabinoids were flow-through analyzed by MRM on a 5500 QTrap triple-quadrupole linear ion trap mass spectrometer equipped with a Turbo V Ion Source (AB SCIEX, Darmstadt, Germany). Ions that were positively and negatively charged were simultaneously analyzed by using the ‘positive-negative-switching’ mode. The MRM transitions monitored for quantification of endocannabinoids in positive ion mode were as follows: AEA, *m*/*z* 348.3 to *m*/*z* 62.3; AEA-d_4_, *m*/*z* 352.3 to *m*/*z* 62.1; 2-AG, *m*/*z* 379.1 to *m*/*z* 287.2; 2-AG-d5, *m*/*z* 384.2 to *m*/*z* 287.2. For the arachidonic acid, the precursor and metabolite of eCBs, the following transitions were used in negative ion mode: AA, *m*/*z* 303.05 to *m*/*z* 259.1; AA-d_8_, *m*/*z* 311.04 to *m*/*z* 267.0. Calibration solutions were prepared using commercially available standards of high purity and spiked with a mixture of deuterated endocannabinoids (Biomol GmbH, Hamburg, Germany). The quantification of endocannabinoids was carried out using Analyst 1.6.3 software, Framingham, MA 01701, U.S.A. The eCBs values were normalized to the protein content of the samples [28,29].

### 2.13. Statistics 

The unpaired *t*-test was used to compare WT with p62 KO mice. Mixed 2-way was applied with genotype and time as between and within groups factors, respectively, followed by Bonferroni’s multiple comparison post hoc test when appropriate. All tests were 2-tailed, and level of significance was set at *p* < 0.05. The statistical analysis and the graphs were made with GraphPad Prism version 7 (GraphPad Software, San Diego, CA, USA). Numerical values were presented as mean ± SEM and n refer to the number of mice.

## 3. Results

### 3.1. Endocannabinoid Levels Are Altered in the Amygdala of Adult p62 KO Mice

The ECS is known to regulate emotion and memory processes [19]. The involvement of the endocannabinoid tone in p62 KO mice was not considered in former studies. To this end, endocannabinoid levels in brain regions were quantified by LC-MS/MS. We decided to measure the AEA, 2-AG, and arachidonic acid (AA) levels on mature 7-months-old male WT (N = 8) and p62 KO (N = 8) mice as neurodegenerative effects were described to be present in mature mice [23]. As areas of interest, the amygdala and hippocampus were selected for purposes of anxiety and memory-related behavior since increased anxiety and a defect in working memory of p62 KO were described priorly [23]. 

First, the obesity phenotype of p62 KO mice was confirmed in this group of animals at the age of 7 months (Figure 1A). The endocannabinoid levels of 2-AG, AEA, and AA measured in the hippocampus were comparable between p62 KO and WT mice (Figure 1B–D). A significant reduction by more than 50% of AEA was detected in the amygdala of p62 KO mice in comparison to its WT littermates (Figure 1E). The amounts of 2-AG and AA were similar between genotypes (Figure 1F,G).

Our findings of a significant reduction of AEA in the amygdala of p62 KO mice compared to WT mice could indicate that p62 KO mice were more likely to be stressed or experience anxiety. Especially in the basolateral amygdala, AEA signaling was associated with the regulation of stress response and anxiety. Former studies revealed that exposure to stress rapidly reduced AEA signaling in the basolateral amygdala [17,30]. In addition, emotional memory consolidation was also shown to involve AEA signaling [31]. Therefore, behavioral testing of p62 KO mice was addressed to further examine anxiety-related behavior and memory function.

### 3.2. Depletion of p62 Does Not Result in Increased Anxiety

The behavior of male p62 KO mice and their WT littermates was studied in 7-months-old animals that were housed in groups. We tested locomotion and exploratory behavior in an open field arena (Figure 2A). The total distance traveled was similar for WT and p62 KO (Figure 2B). Time spent by mice in the center of the arena was used as an indicator of fear and exploration behavior, as mice naturally tend to avoid open spaces. No differences were found between p62 KO and WT mice (Figure 2C).

To further investigate exploratory and fear behaviors, we used the elevated plus maze (Figure 2D). Therefore, mice were introduced to the center of the elevated plus maze facing an open arm. The total distance traveled was significantly longer in p62 KO mice than in their WT littermates (Figure 2E), while the time spent in the open arms of the arena was comparable between p62 KO and WT mice (Figure 2F).

Next, we performed the light–dark box test (Figure 2G) in group-housed mice to examine anxiety and exploratory behavior. The distance traveled was similar in WT and p62 KO mice (Figure 2H). Time to re-enter the light arena was used as an indicator of anxiety and exploratory drive. Both WT and p62 KO mice re-entered the light zone after approximately 50 s (Figure 2I). The time spent by mice in the illuminated part of the arena was similar between genotypes (Figure 2J).

Next, we examined whether individually housed animals exhibited modulated behavior in the open field or elevated plus maze. These male animals were 3 to 4 months old and moved a similar total distance in the open field (Figure 2K). Our results revealed that mice with a deletion of p62 showed a slight tendency to spend more time in the center of the arena compared to WT littermates (Figure 2L), indicating reduced anxiety-like behavior. However, in the elevated-plus maze, the total distance traveled (Figure 2M) and time spent in the open arms of the arena were comparable between p62 KO and WT mice in single caged animals (Figure 2N).

Taken together, we identified in p62 KO animals a tendency to spend more time in the middle of the open field, which could be interpreted as reduced anxiety or increased exploratory behavior in mice kept in single cages. Furthermore, the significant increase in distance traveled in the elevated plus-maze in mice housed in groups also suggests increased exploratory drive. Spontaneous locomotion, exploratory, and fear behaviors of p62 KO mice were predominantly comparable to those of their WT littermates in adult and in more mature mice. Thus, p62 KO mice showed in our hands a slight tendency towards a novelty-seeking phenotype that was independent of age and type of housing, whereas we could not detect a strong anxiety behavior as previously described [23].

### 3.3. Intact Working Memory and Increased Exploratory Behavior in p62 KO Mice

In a Y-maze, we tested spontaneous alternation and spatial working memory of the individually housed mice by using their motivation to explore a new environment (Figure 3A). The same group of mice that were housed individually in single cages was used for all following experiments. For alternation behavior, mice had to discriminate a new arm of the maze from a previously visited arm. Both WT and p62 KO mice showed similar alternation behavior (Figure 3B). Both genotypes entered a comparable number of arms during the test period (Figure 3C). The distance traveled was not different in WT and p62 KO mice (Figure 3D), suggesting normal locomotion and working memory abilities of p62 KO mice.

To test novel object recognition, mice were habituated to the open field arena for two days (Figure 3E). Our results of the first open field are shown in Figure 2M,L. The mice were introduced to the open field arena a second time, and the distance traveled was analyzed. The mice traveled a comparable distance regardless of genotype (Figure 3F). We analyzed habituation to the arena by comparing the distance traveled in the first open field with the distance traveled in the second open field. WT and KO mice showed lower activity in the second open field, as expected (Figure 3G). In the next round of experiments, two identical objects were presented (Figure 3H). p62 KO mice spent significantly more time exploring the objects (Figure 3I) and moved a longer distance than their WT littermates, especially at the beginning of the experiment resulting in a significant interaction (F _(9,198)_ = 2.5, *p* = 0.01) (Figure 3J). The following day, a familiar and a novel object were introduced. Normally, the mice remembered the familiar object and distinguished it from the new object. Since the exploration time for the new object was similar to that for the familiar one in both genotypes (WT, 54% ± 3%, N = 12; KO, 55% ± 4%, N = 12; *p* = 0.72), we cannot draw a conclusion about the object recognition and long-term retrieval behavior of our p62 KO mice. Overall, we detected an intact working memory and an increased exploratory behavior of p62 KO mice.

### 3.4. P62 KO Mice Display Normal Contextual, Cued, and Long-Term Fear Memory and Showed Reduced Freezing Behavior

Fear conditioning was used as a behavioral paradigm to test the long-term and contextual fear memory of p62 KO and WT mice at a delay of 24 h and 7 days after the conditioning phase (Figure 4A). During our conditioning phase, both genotypes responded as expected with increased freezing behavior to the repetition of foot shocks, confirming the efficacy of the fear conditioning protocol (Figure 4B: time F_(2,44)_ = 65, *p* < 0.001). We observed no difference between WT and p62 KO mice (Figure 4B: genotype F_(1,22)_ = 2.1, *p* = 0.16; interaction F_(2,44)_ = 1.2, *p* = 0.31). The next day (24 h), we placed the mice in the same context where they had received the foot shocks. p62 KO mice showed a slight.

Reduction in freezing time of less than 10% in the first minute compared to their WT littermates (Figure 4C). Overall, WT and p62 KO mice showed similar levels of immobility, confirming an intact contextual fear memory at 24 h (Figure 4C: genotype F_(1,22)_ = 0.32, *p* = 0.58; interaction F_(4,88)_ = 0.89, *p* = 0.48; time F_(4,88)_ = 4.2, *p* = 0.004). Two hours after the contextual memory test, we tested the cued memory. The first two minutes of freezing were used to display the baseline freezing response before applying the conditioning tone. WT and p62 KO mice responded to the tone with increased freezing that decreased with time (Figure 4D: time F_(5,110)_ = 100, *p* < 0.001). However, the freezing response to the conditioning tone differed slightly between genotypes, as p62 KO mice showed a decrease in freezing during the first minute (minute 3) of tone presentation (Figure 4D: interaction F_(5,110)_ = 3.6, *p* = 0.004). Both genotypes responded with increased immobility to the tone, confirming intact cued memory in p62 KO and WT mice (Figure 4D: genotype F_(1,22)_ = 1.7, *p* = 0.20). We examined contextual memory a second time after 7 days. After 24 h, p62 KO mice showed reduced freezing behavior only during the first minute after the tone (Figure 4E). Overall, p62 KO and WT mice were able to retrieve contextual fear memory for a duration of 7 days (Figure 4E: genotype F_(1,22)_ = 1.9, *p* = 0.19: interaction F_(4,88)_ = 2, *p* = 0.11: time F_(4,88)_ = 3.2, *p* = 0.02). Similarly, cued memory was intact in p62 KO mice after 7 days (Figure 4F: genotype F_(1,22)_ = 6.1, *p* = 0.02: interaction F_(5,110)_ = 1.6, *p* = 0.16: time F_(5,110)_ = 71, *p* < 0.001.

Taken together, our results indicate that the contextual and cognitive fear memory of p62 KO mice was intact and memory remained intact for at least 7 days. Of note, we detected that p62 KO mice were more mobile compared to their WT littermates during the conditioning phase (Figure 4G) but not during the context (Figure 4H) or cued phase (Figure 4I), suggesting a potentially increased exploratory behavior.

### 3.5. P62 KO Mice Display Normal Motor Learning, Coordination, Strength, and Endurance

To investigate whether the deletion of p62 is important for motor functions, we performed the rotarod and pole test. The rotarod test was used to study coordination, strength, and endurance. The mice require all of these characteristics to walk on the rotating rod (Figure 5A). Both, WT and p62 KO mice, improved over time (Figure 5B: time F_(11,242)_ = 22, *p* < 0.001). Although p62 KO mice performed slightly worse than their WT littermates, the difference was not statistically significant (Figure 5B: genotype F_(1,22)_ = 0.14, *p* = 0.71; interaction F_(11,242)_ = 1.1, *p* = 0.39). The pole test is commonly used to study movement disorders associated with the basal ganglia [32] and was selected because of the high expression of the CB1 receptor in this brain area (Figure 5C). WT and p62 KO mice improved over time as they were able to turn closer to the top of the pole (Figure 5D: time F_(2,42)_ = 4.9, p < 0.01). Both WT and p62 KO mice turned near stage 2, with no difference between genotypes and no impaired basal ganglia-based motor behavior (Figure 5D: genotype F_(1,22)_ = 2.9, *p* = 0.10; interaction F_(2,44)_ = 0.45, *p* = 0.64). The time mice needed to reach their home cage also improved with time (Figure 5E: time F_(2,42)_ = 7.3, *p* < 0.002) and was comparable between genotypes (Figure 5E: genotype F_(1,21)_ = 0.28, *p* = 0.60; interaction F_(2,42)_ = 1, *p* = 0.38).

Our results of the rotarod and pole tests showed that 3–4-months-old WT and p62 KO mice were able to learn and improve motor tasks. Both genotypes were able to coordinate complex movements through coordination, strength, and endurance.

### 3.6. Less THC Induced Effect on Hypolocomotion in p62 KO Compared to WT Mice

The activation of the ECS by agonists such as THC induces hypolocomotion and hypothermia in mice [33,34]. Since the p62 KO mice in our experiments showed a slightly increased explorative drive and altered freezing behavior, although the measured reduced AEA levels should lead to an anxiogenic behavior, we now aimed to investigate the effect of THC on locomotion in the open field. With this experiment, we aimed to study if the ECS-dependent responses were affected in p62 KO mice. For this purpose, we used 7–8-month-old p62 KO and WT females, which showed a slightly increased weight in the KO animals (Figure 6A). We decided to use female mice for this experiment because females showed stronger responses to THC and other cannabinoids [35,36,37]. THC induces a hypothermic response that was observed in both genotypes and was not different between WT and KO animals (Figure 6B).

The locomotor activity of the mice was analyzed 30 min after THC injection in the open field test. Administration of THC resulted in a significantly greater reduction in locomotion in p62-WT mice (Figure 6C) than in p62 KO littermates determined by distance travelled (Figure 6C), indicating a kind of resilience of p62 KO mice to THC. No difference in time spent in the middle zone of the arena was observed (Figure 6D).

Taken together, p62 KO mice were able to respond to THC because they exhibited hypothermia but showed significantly less reduction in locomotor activity 30 min after THC application, indicating a reduced function of the ECS due to loss of p62.

## 4. Discussion

The ECS has been implicated in both anxiety-related behavior as well as memory and learning [17,18]. AEA signaling is particularly associated with the basolateral amygdala [19], the brain area that is involved in the regulation of stress response, anxiety, and exploratory behavior [38].

Our results showed for the first time that mice with a deletion of the protein p62 have differences in their endocannabinoid levels. The decreased AEA level in the amygdala of mature p62 KO mice (7 months, group housed) may be associated with the slight exploratory novelty-seeking behavior determined by more locomotor activity and the reduced freezing responses of the mice. These results imply a contrary effect of the ECS in p62 KO mice that was further supported by the relatively low effect of THC administration on the locomotor activity in p62 KO in comparison to WT littermates. Of note, we have not detected general differences between motor function and working memory.

Mice with a deletion of p62 developed an age-dependent obesity phenotype [22,39], which we confirmed in our mouse line. Despite the high body weight, the mice showed increased activity in the experiments performed. The animals appeared more curious overall and showed slightly reduced freezing behavior in the cued fear conditioning test. It must be considered that an expression of increased locomotor activity could also be interpreted as an increased exploratory drive. However, hyperlocomotor activity can be excluded as animals in the open field did not show increased activity with respect to distance moved compared with their WT littermates.

The decreased levels of AEA in the amygdala and the reduced effect of THC on locomotion in the p62 KO animals may suggest that the absence of p62 leads to a dysfunction of the ECS. According to current knowledge, the endocannabinoid levels are controlled by the activities of the enzymes responsible for the biosynthesis and degradation [40]. Possibly, p62, a protein that serves as a receptor on autophagy cargos, is also involved in the control of endocannabinoid levels which could occur by regulating the metabolic enzyme degradation. Of note, enhanced autophagy was identified in old CB1 KO animals [3]. Therefore, a regulatory effect of p62 on CB1 receptor protein levels through protein degradation or internalization [41,42] may be possible, which could explain the different responses of p62 KO and WT animals after THC administration on locomotor activity. Yet, the effect of THC on hypothermia did not differ between the two genotypes, suggesting that there is no general change in CB1 receptor protein levels. However, this relationship needs to be investigated in further detailed analyses.

The high complexity of the regulatory effects of the ECS on behavioral processes makes it virtually impossible to predict the expected results. This is especially true when investigating emotional behavior since the pharmacological increase in AEA levels might induce opposite effects on anxiety behavior depending on the emotional status of the animals [17]. Both positive and negative effects of altered cannabinoid receptor signaling have also been described for memory formation, which can be influenced by different agonist or antagonist treatments depending on the application, dosage, and behavioral paradigm [18,43]. For exogenous cannabinoid administrations of CB1 receptor agonists, it is well known to have a biphasic dose-dependent effect on anxiety with anxiolytic effects of low and anxiogenic effects of high doses [19]. The anxiolytic effect of low CB1 agonist administration is mediated by glutamatergic CB1 receptors, whereas GABAergic CB1 receptors are responsible for the anxiogenic effect of high CB1 agonist doses [44]. The specific experiments of mice with endogenously low AEA levels have not been performed yet, however, increased anxiety response would be predicted [19] since pharmacological experiments with an inhibitor of the AEA synthesizing enzyme *N*-acylphosphatidylethanolamine phospholipase D (NAPE-PLD) LEI-401 in mice have resulted in decreased brain AEA levels and led to an increased fear extinction response in mice [45]. In line with that, in FAAH KO animals with elevated AEA levels a decreased anxiety phenotype was described [46]. The inhibition of AEA uptake/degradation reduced freezing, whereas the inhibition of 2-AG degradation promoted freezing, which was similar to the effects of a CB1 agonist and dependent on CB1 expressed by GABAergic neurons [47]. Of note also, the CB2 receptor had been shown to be involved in freezing responses in mice since a blockade of CB2 by AM630 caused a slight decrease in fear [47].

The ECS is of particular importance for the forgetting of negative experiences that is investigated by a specific protocol of the fear conditioning, the so-called fear extinction experiment [48]. In this test, a previously learned Pavlovian conditioning between an acoustic stimulus and a painful electrical shock is “disassociated” so that it can be determined how well animals learn to separate the harmless stimulus from the negative experience. In our experiments, we aimed to investigate whether the p62 KO mice were worse at learning and remembering (cued and context memory) compared to their WT littermates, but not extinction. Our hypothesis was based on the fact that a neurodegenerative phenotype similar to Alzheimer’s disease had been described by others for p62 KO animals [23].

To our knowledge, only one other publication has examined the behavior of p62 KO mice describing an anxiogenic behavior of young (2 months old) p62 KO mice, as well as the reduced working memory of young and mature (6 months old) p62 KO mice accompanied by increased accumulation of ubiquitinated tau protein in the brain of mature p62 KO mice [23]. Of note, a limitation of this former work is an unusual behavior that was present in WT animals, as they showed no favor to dark places but visited the open arms in an elevated plus maze with the same preference as the closed arms. In comparison to the WT animals, the p62 KO mice behaved as expected and preferred the dark area in that former work [23]. This result led to the conclusion by the authors that p62 knockouts are more anxious [23]. Therefore, we conclude that the previously published result of an anxiogenic phenotype in p62 KO animals [23] was not representative and that our data now demonstrate that p62 KO animals showed increased activity and curiosity independent of the age and the housing condition. Repetitive handling of mice prior to behavioral testing improves spatial cognition and reduces anxiety-like behavior in mice [49]. For all behavior experiments carried out in this work, mice were handled to become familiar with the experimenter and to reduce the stress of mice during behavioral testing. Therefore, handling or the absence of mice handling could have impacted the behavioral output [49], which could result into different findings. Furthermore, the genetic background of mice has an impact on the phenotype of knockout mice [50]. Even though this work and the publication by Ramesh Babu et al. [23] used C57BL/6 mice, the sub-strain and the time of breeding without backcrossing to the original sub-strain potentially impact the phenotype of mice. In this study, we used C57BL/6J mice for the backcrossing of p62 KO mice. A study by Matsuo and colleagues showed that even small genetic differences between C57BL/6J, C57BL/6N and C57BL/6C influenced behavioral and, specifically anxiety phenotypes [51].

Based on former publications [23] together with the low AEA concentrations in the p62 KO amygdala, we found, against our expectations, signs of less freezing and a slight trend toward increased exploratory behavior in p62 KO animals. In addition, the mice showed increased locomotor activity in several experiments, even after THC administration. Our results may indicate that loss of p62 is involved in the regulation of CB1-mediated functions. Whether this functional modulation is processed by autophagy [3] and whether GABA- or glutamatergic CB1 receptor signaling involved in biphasic cannabinoid responses is differentially affected by loss of p62 may be a direction for future investigation. In addition, future experiments focusing on stress-dependent endocannabinoid levels and the effects of CB1 receptor ligands in p62 KO animals may help to clarify these issues.

## Figures and Tables

**Figure 1 cells-11-01517-f001:**
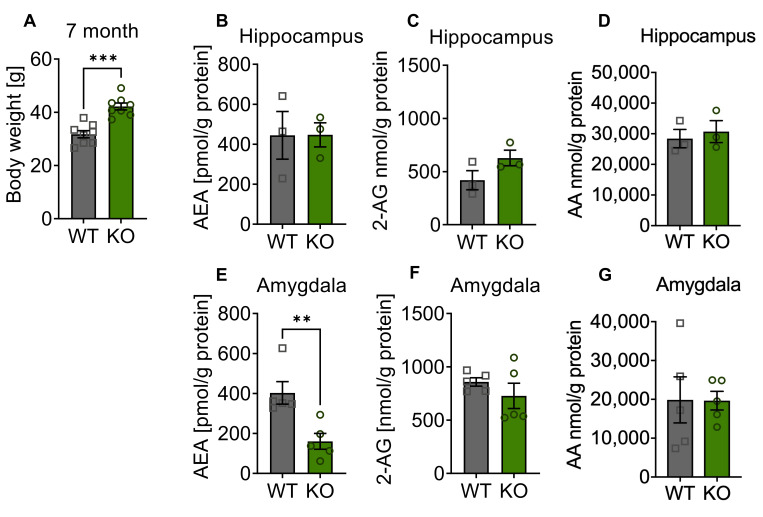
Altered endocannabinoid levels in the amygdala of mature p62 KO mice. (**A**) Body weight of p62 KO male mice aged 7 months was significantly increased compared to WT mice. (**B**–**D**) Endocannabinoid measurement in the hippocampus revealed no difference in level of 2-AG, AEA, or AA between genotypes. (**E**) AEA was significantly decreased in p62 KO mice in the amygdala. (**F**,**G**) Levels of 2-AG and AA in the amygdala were comparable between genotypes. Squares and circles represent single replicates. ** *p *< 0.01, *** *p *< 0.001 (unpaired *t*-test). WT male N = 8; KO male N = 8.

**Figure 2 cells-11-01517-f002:**
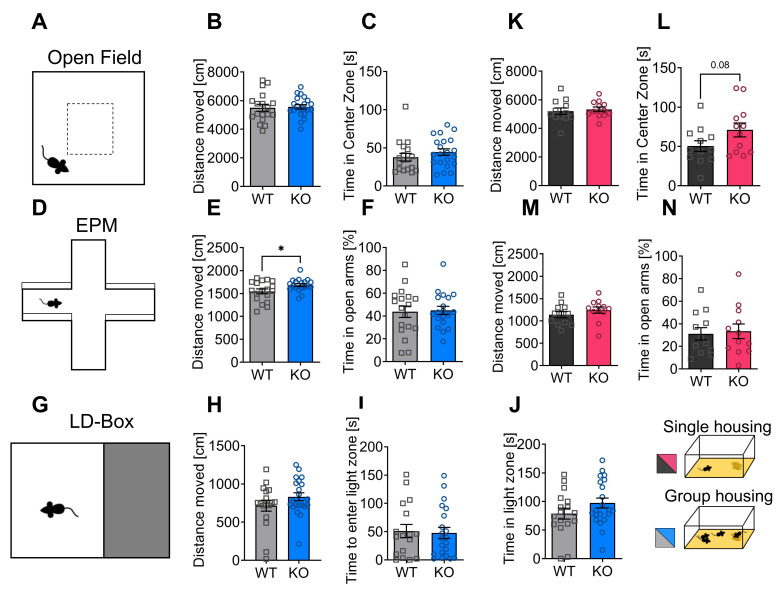
Depletion of p62 does not result in increased anxiety. (**A**) Illustration of the open field arena and the center zone (20 cm × 20 cm). (**B**) Both genotypes moved an equal distance in the open field arena. (**C**) The time p62 KO and WT mice spent in the center was similar. (**D**) Schematic view of elevated plus maze. (**E**) p62 KO mice moved a significantly longer distance in the elevated plus maze compared to WT mice. (**F**) The time spent in the open arms of the elevated plus maze was almost identical. (**G**) Schematic illustration of the light–dark box test. (**H**) Distance moved in the light zone of the arena was similar in WT and KO mice. (**I**) Time mice needed to re-enter the light arena was almost identical. (**J**) The time mice spent in the light arena was comparable. (**K**) WT and p62 KO mice moved a comparable distance in the open field. (**L**) p62 KO mice spent slightly more time in the center of the open field than their WT littermates. (**M**) WT and p62 KO mice moved a comparable distance in the elevated plus maze. (**N**) Time spent in the open arms of the elevated plus maze was comparable between genotypes. * *p* < 0.05 (unpaired *t*-test). Squares and circles represent single replicates. All error bars represent ± SEM. Group 1 (B-J group housing, blue): 3–4 months old, male; WT = 12, KO = 12. Group 2 (K-N single housing, red): 7 months, male: WT N = 18, KO N = 21. EPM, elevated plus maze; LD-Box, light–dark box.

**Figure 3 cells-11-01517-f003:**
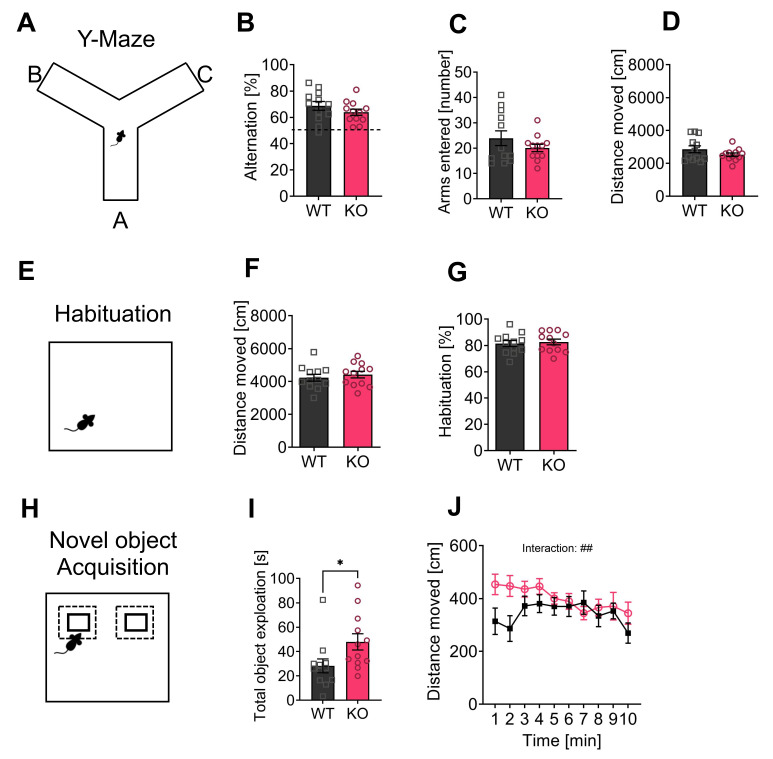
Working memory and object recognition are intact in p62 KO mice. (**A**) Representative image of Y-maze arena. (**B**) WT and p62 KO mice showed similar alternating behavior. (**C**) Both genotypes entered the arms of the Y-maze at a comparable number and (**D**) moved a similar distance. (**E**) Representative image of the open field arena. Mice were placed a second time to the same arena to analyze habituation behavior. (**F**) WT and p62 KO mice moved a comparable distance, and (**G**) showed intact habituation behavior. (**H**) Representative image of the novel object set up showing two identical objects that were presented. (**I**,**J**) Two identical objects were presented during the exploration phase and p62 KO mice spent a significantly longer time exploring the objects (**I**) and moved a longer distance (**J**) compared to WT mice. * *p* < 0.05. Data were analyzed by unpaired *t*-test or 2-way ANOVA repeated measurements. Squares and circles represent single replicates. All error bars show mean ± SEM. Age: 3–4 months old, WT male N = 12; KO male N = 12.

**Figure 4 cells-11-01517-f004:**
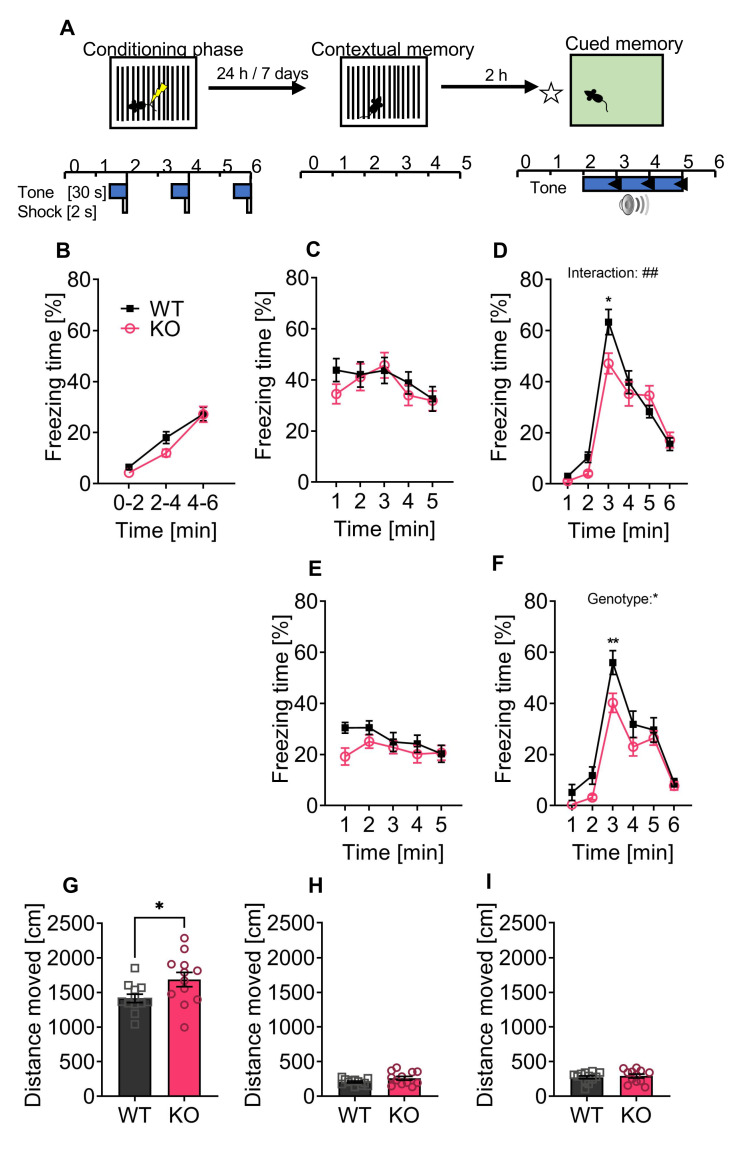
p62 KO mice display normal contextual, cued, and long-term fear memory and showed a reduced freezing after the acoustic stimulus. (**A**) Schematic representation of the fear conditioning protocol. (**B**) During the conditioning phase a foot shock was paired with a tone and repeated three times. Freezing response to the conditioning protocol was similar between p62 KO and WT mice and increased with time and foot shock repetition. (**C**) Contextual memory was tested after 24 h following conditioning. WT and p62 KO mice were introduced to the same arena where they have received foot shocks and showed increased but comparable freezing response. (**D**) Cued memory was tested using a different environment and presentation of the paired tone. The freezing response during the first two minutes before applying the tone was used for the baseline response. p62 KO and WT mice showed intact cued memory with a reduced freezing response in p62 KO animals in the first minute of the tone. (**E**) Contextual memory of mice was tested again after 7 days and confirmed intact contextual memory of WT and p62 KO mice. (**F**) Cued memory tested after 7 days showed intact memory of p62 KO and WT mice with again a reduced response in p62 KO mice when the tone was applied. (**G**) During the conditioning phase p62 KO animals moved a significantly longer distance than the WT mice. No difference in distance moved was identified during (**H**) the context and the (**I**) cued phases. * *p* < 0.05; ** *p* < 0.01. Data were analyzed by using 2-way ANOVA repeated measurements (**B**–**F**) or by using unpaired *t*-test (**G**–**I**). All error bars show mean ± SEM. Squares and circles represent single replicates. WT male N = 12; KO male N = 12.

**Figure 5 cells-11-01517-f005:**
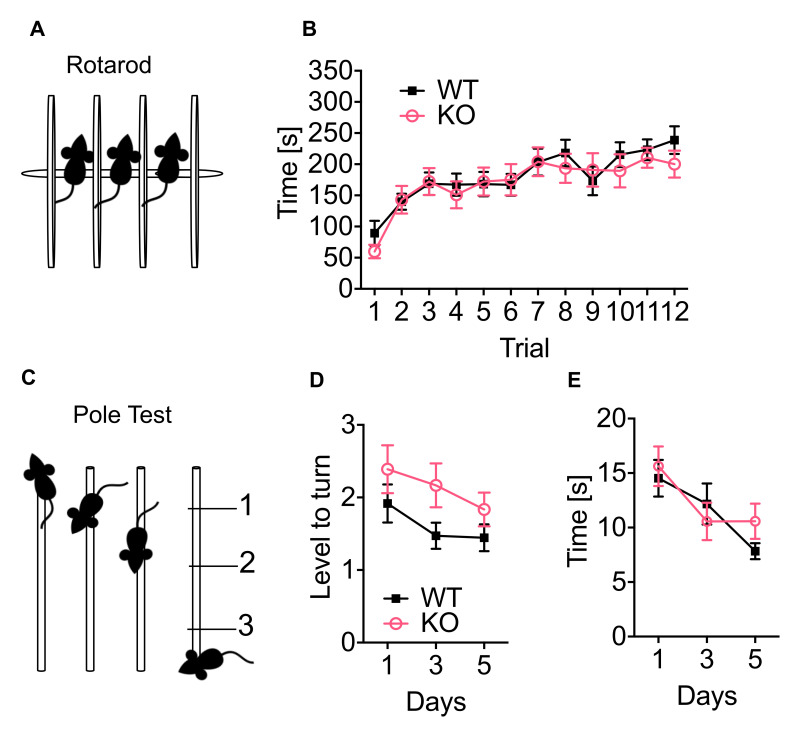
p62 KO mice show normal motor learning, coordination, strength, and endurance. (**A**) Representative image of the rotarod apparatus. (**B**) The time mice spent running on the rotarod was improved during training and was similar between p62 KO and WT mice. (**C**) Representative image of the pole test. (**D**) The level to turn (180°) was improved over time and was comparable between p62 KO and their WT littermates. (**E**) The time mice needed to climb down the pole was improved over time and equally in p62 KO and WT mice. Data were analyzed by using 2-way ANOVA repeated measurements. All error bars show mean ± SEM. Age: 3–4 months old, male WT N = 12; KO N = 12.

**Figure 6 cells-11-01517-f006:**
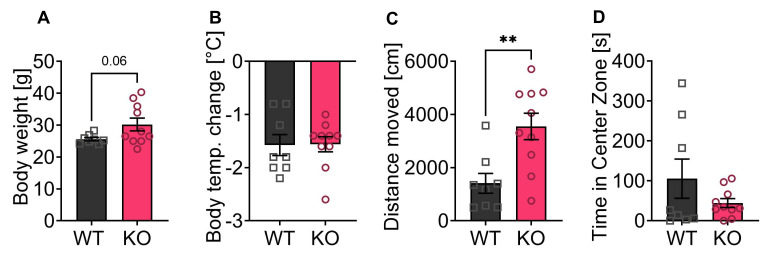
Reduced locomotor effects of THC administration in p62 KO mice. (**A**) Body weight of p62 KO mice was slightly increased compared to WT mice (*p* = 0.06). (**B**) Body temperature 30 min after THC injection showed no significant difference between genotypes. (**C**) The distance traveled in open field 30 min after injection was significantly reduced in WT mice compared to p62 KO littermates (vehicle and THC 5 mg/kg). (**D**) The distance traveled in the center of the open field arena 30 min after THC injection was comparable between genotypes (THC 5 mg/kg). ** *p* < 0.01 (unpaired *t*-test). All error bars show mean ± SEM. Squares and circles represent single replicates. Age: 7–8 months old, female, WT N = 8; KO N = 10.

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
