# Peer review of "Behavioral Studies of p62 KO Animals with Implications of a Modulated Function of the Endocannabinoid System"

_cells, 2022, doi:10.3390/cells11091517_

Round 1

Reviewer 1 Report

The paper by Keller et al. shows that P62-deficient mice show a reduced level of anandamide (but not of 2-arachidonoylglycerole) in the amygdala (but not in the hippocampus). One might expect that the knockout mice show increased anxiety but, paradoxically, three behavioural parameters, 1. reduced freezing, 2. increased exploratory behaviour and 3. reduced inhibitory effect of delta9-tetrahydrocannabinol on locomotor activity were altered in the opposite direction.

The manuscript is sound and the results are shown in an accurate manner. I have two questions/suggestions:

  1. Authors have worked with at least three different groups of mice: male – 3-4 months; male – 7 months; female – 7-8 months. It is not clear to me why these three groups were used. Some more information would be helpful.
  2. There are extreme discrepancies to papers by other authors. I am wondering whether these differences may be related to the genetic background of the mice. I remember that sometimes knockout mice with C57BL/6J and C57BL/6N background differ markedly.

A series of minor points should be considered:  

  • 32: help to gain
  • 45-48: I am wondering whether these few lines should be re-phrased since extremely different results were obtained in the studies (7-9).
  • 51: Why do you say FURTHER anxiolytic effects?
  • 62-64: This sentence reminds me of the statement of A. Zimmer that the ECS is a homeostatic system.
  • 75-76: Alzheimer’s disease?
  • 106: Where is KOMP based?
  • 111: One x is missing.
  • 112: ad libitum
  • 127: …spaces; when…
  • 135,154, 177, 220, 346 and many other sites: spent
  • 180: one kind of object…
  • 208: stimulus
  • 210: through
  • 215: jumping off
  • 247: was performed
  • 248: snap frozen
  • 257: The abbreviation BCA should be explained.
  • 258ff: Use sub- and superscripts.
  • 271: deuterated 2-AG???
  • 278: A section related to the sources of drugs is missing.
  • 291: 7-month-old. This is a frequent problem in the manuscript: It should read: Mice were 7 months old (month is a substantive) or Studies were carried out with 7-month-old mice (month is part of an adjective).
  • 297: Body weights are available from Fig. 1A and should not be repeated in the text. By the way: p<0.001!!!!!
  • 300: to their WT…
  • 301-302; 323; 326; 331; 333 and many other sites: See comment to 297.
  • 312: “mature” would perhaps be better than “adult”.
  • 316: decreased!!!!!
  • 319: Perhaps I have read the text not carefully but I missed two important details given in Fig. 2, i.e. that some mice were 3-4 months old and that in part of the experiments group housing and in the rest single housing was studied. This is more important than repetition of values available from Fig. 2.
  • 344: EPM and LD should be explained in the legend.
  • 379: In this line and elsewhere (e.g. Fig. 3, both figure and legend) both alterate and alternate are used. Please use the correct one.
  • 444: Like (instead of Similar to)?
  • 472: significantly longer
  • 489: The time mice needed to reach their home cage…
  • 507: of CB1 receptors (instead of the ECS)
  • 510: reduced (instead of altered)
  • 528: Delete “was normalized to baseline and”. I guess that baseline temperature was identical in WT and KO mice?
  • 538: The sentence is incomplete.
  • 552: interpreted
  • 575: I am wondering whether one should also quote the work by Bilkei-Gorzo et al. who showed that cognitive abilities were increased in young adult and decreased in mature CB1 receptor KO mice.
  • Figure 4 and 5: I would suggest to arrange the two figures in a manner that the figure itself and the legend in full length appear on one page.

Author Response

We would like to thank our reviewers for their thorough review of our manuscript and their critical but constructive comments. Please find attached the revised version of our manuscript (automatically named author-coverletter) with the main changes marked in yellow and below our responses to your comments.

1. Authors have worked with at least three different groups of mice: male – 3-4 months; male – 7 months; female – 7-8 months. It is not clear to me why these three groups were used. Some more information would be helpful.

For the behavioral experiments, we used groups of younger and older adult animals, as there was evidence from previous publications that there are age-dependent changes in the brains of p62 KO animals (23). Together with the age-dependent obesity phenotype we wanted to exclude that these phenotypes have an influence on the behavioral results. As our primary results in the 7-month-old group did not match our expectations, we wanted to exclude that the age affects the results. Thus, we additionally used the younger animals. Concerning the THC experiment, we used female mice because they produce stronger responses to cannabinoids (35-37). We used the females in the more mature age because we identified the altered AEA levels in the older animals.

We included an explanation in the manuscript (page 13, line 773).

35. Fattore, L., and Fratta, W. (2010) How important are sex differences in cannabinoid action? Br J Pharmacol 160, 544-548

36. Patton, G. C., Coffey, C., Carlin, J. B., Degenhardt, L., Lynskey, M., and Hall, W. (2002) Cannabis use and mental health in young people: cohort study. BMJ 325, 1195-1198

37. Tseng, A. H., and Craft, R. M. (2004) CB(1) receptor mediation of cannabinoid behavioral effects in male and female rats. Psychopharmacology (Berl) 172, 25-30

2. There are extreme discrepancies to papers by other authors. I am wondering whether these differences may be related to the genetic background of the mice. I remember that sometimes knockout mice with C57BL/6J and C57BL/6N background differ markedly.

In part the discrepancies may result by different mouse strains used in both studies. We included a paragraph stating that the handling and the genetic background of mice may influence behavioral studies (page 15, lines 903-920).

But there are additional reasons for the discrepancies to previously published work. In particular, there is a limitation of the published study by Ramesh-Babu et al. in which an unusual behavior was present in WT animals in the elevated plus maze where they did not show a preference for dark places but visited the open arms with a similar preference to the closed arms. In comparison to the WT animals, the p62 KO mice behaved as expected and preferred the dark area in that former work (23). This unusual behavior of the WT mice (not p62 KO mice) led to the conclusion by the authors that p62 KO are more anxious. In addition, instead of determining time spent in the center of the open field, the authors assessed anxiety-like behavior by counting fecal deposits after 10 min in the open field maze and found that p62 KO mice had an average of 2.8 fecal deposits, compared to 1.3 fecal deposits of WT mice (23).

We conclude that the previously published result of an anxiogenic phenotype in p62 KO animals was not representative. We include this explanation in our manuscript (page 15, lines 589 – 606).

23. Ramesh Babu, J., Lamar Seibenhener, M., Peng, J., Strom, A. L., Kemppainen, R., Cox, N., Zhu, H., Wooten, M. C., Diaz-Meco, M. T., Moscat, J., and Wooten, M. W. (2008) Genetic inactivation of p62 leads to accumulation of hyperphosphorylated tau and neurodegeneration. J Neurochem 106, 107-120

3. A series of minor points should be considered:  

Thanks for all your minor suggestions and comments that we now changed in our revised manuscript.

Reviewer 2 Report

Manuscript ID cells-1666058

Title: Behavioral studies of p62 KO animals with implications of a modulated function of the endocannabinoid system

Dear Authors,

This paper is interesting, but it requires major revisions.

I have some recommendations to improve the quality of this paper:

1, Some typos are present throughout the text. I noted very long sentences throughout the manuscript.

2, Line 304-310, the font size is different.

3, Figure should label properly and consistently. The scale bar of the first figure is not consistent for Endocannabinoid measurement. Sometimes, it can be difficult to organize a steady scale bar but here grouping of similar candidates would be helpful for a constant scale bar. The green bar of section B possessed less value as compared to the green bar of E but apparently, the height shows otherwise. For grouping, use B and E, C and F (like this section), and D and G.     

4, The same method should apply to other figures.

5, Please ensure that all the abbreviations have been defined in parentheses the first time they appear in the main text.

6, The authors should read the manuscript thoroughly and check: a) space between words; b) punctuation; c) English of some sentences.

7, Keywords are not appropriate, Similar words are used in the title and abstract that limit the importance of present keywords. These should be changed.

8, References: Please cite more recent original papers.

In conclusion, this article has the potential to be of high interest to Behavioral studies of p62 KO animals with implications of a modulated function of the endocannabinoid system readership. However, a major revision is required.

Author Response

We would like to thank our reviewers for their thorough review of our manuscript and their critical but constructive comments. Please find attached the revised version of our manuscript (automatically named author-coverletter) with the main changes marked in yellow and below our responses to your comments.

1, Some typos are present throughout the text. I noted very long sentences throughout the manuscript.

We eliminated typos and all mean-values +/- SEM and p-values throughout the manuscript if they are represented in a figure. This shortens most of the sentences dramatically.

2, Line 304-310, the font size is different.

We changed it.

3, Figure should label properly and consistently. The scale bar of the first figure is not consistent for Endocannabinoid measurement. Sometimes, it can be difficult to organize a steady scale bar but here grouping of similar candidates would be helpful for a constant scale bar. The green bar of section B possessed less value as compared to the green bar of E but apparently, the height shows otherwise. For grouping, use B and E, C and F (like this section), and D and G.     

and 4, The same method should apply to other figures.

The figures were adjusted in their scaling and Figure 1 was also changed in the order of the graphs.

5, Please ensure that all the abbreviations have been defined in parentheses the first time they appear in the main text.

We identified some abbreviations in the text that were not explained before.

6, The authors should read the manuscript thoroughly and check: a) space between words; b) punctuation; c) English of some sentences.

We have made corrections throughout the text.

7, Keywords are not appropriate, Similar words are used in the title and abstract that limit the importance of present keywords. These should be changed.

We changed the keywords.

8, References: Please cite more recent original papers.

We included more recent publications (page 2, lines 54-55 and 60-62).

Round 2

Reviewer 2 Report

The author has addressed all the raised concerns. Best regards